# Pathophysiology of Bone Loss in Patients with Prostate Cancer Receiving Androgen-Deprivation Therapy and Lifestyle Modifications for the Management of Bone Health: A Comprehensive Review

**DOI:** 10.3390/cancers12061529

**Published:** 2020-06-10

**Authors:** Tae Jin Kim, Kyo Chul Koo

**Affiliations:** 1Department of Urology, CHA Bundang Medical Center, CHA University College of Medicine, Seongnam 13496, Korea; tjkim81@cha.ac.kr; 2Department of Urology, Gangnam Severance Hospital, Yonsei University College of Medicine, Seoul 06229, Korea

**Keywords:** androgen, bone diseases, endocrine, lifestyle, prostatic neoplasm, quality of life

## Abstract

Androgen-deprivation therapy (ADT) is a systemic therapy administered for the management of advanced prostate cancer (PCa). Although ADT may improve survival, long-term use reduces bone mass density (BMD), posing an increased risk of fracture. Considering the long natural history of PCa, it is essential to preserve bone health and quality-of-life in patients on long-term ADT. As an alternative to pharmacological interventions targeted at preserving BMD, current evidence recommends lifestyle modifications, including individualized exercise and nutritional interventions. Exercise interventions include resistance training, aerobic exercise, and weight-bearing impact exercise, and have shown efficacy in preserving BMD. At the same time, it is important to take into account that PCa is a progressive and debilitating disease in which a substantial proportion of patients on long-term ADT are older individuals who harbor axial bone metastases. Smoking cessation and limited alcohol consumption are commonly recommended lifestyle measures in patients receiving ADT. Contemporary guidelines regarding lifestyle modifications vary by country, organization, and expert opinion. This comprehensive review will provide an evidence-based, updated summary of lifestyle interventions that could be implemented to preserve bone health and maintain quality-of-life throughout the disease course of PCa.

## 1. Introduction

Prostate cancer (PCa) is the second most commonly diagnosed of all male cancers worldwide and is the fifth leading cause of cancer death in men [1]. Treatment options for PCa patients include surgical procedures, radiation therapy, androgen-deprivation therapy (ADT), and chemotherapy, depending on the stage and course of the disease. Androgen receptor (AR) signaling actively promotes growth, proliferation, and invasiveness of PCa. ADT is most commonly administered as a gonadotrophin-releasing hormone (GnRH) analog, which induces downregulation of the pituitary-gonadal axis and subsequent suppression of the testicular production of testosterone, resulting in chemical castration or the reduction in androgen activity to castration levels. ADT is a standard therapy for patients with aggressive and systematic disease, in which the suppression of androgen activity to castration levels reduces disease-related morbidity and prolongs survival [2].

PCa is an androgen-dependent disease; therefore, ADT is a keystone in the treatment for hormone-sensitive metastatic or advanced disease [3]. Docetaxel chemotherapy and androgen receptor axis-targeted agents, including abiraterone or enzalutamide, have shown clinical efficacy in patients with castration-resistant PCa (CRPC) [4]. Still, ADT is the mainstay treatment for patients with advanced disease or CRPC and should be continuously maintained [4]. Despite the oncological benefits of ADT, long-term administration induces clinical sequelae of reduced quality of life (QoL), sexual dysfunction, and adverse metabolic effects, such as the development of insulin resistance, reduced muscle mass, increased fat mass, and loss of bone mineral density (BMD) [3,4,5,6]. The decline in BMD is a devastating adverse effect of ADT that increases the risks of skeletal-related events (SREs) such as pathological fractures and spinal cord compression. The need for bone surgery or adjunctive treatments for these patients is known to reduce QoL and survival [7].

Loss of BMD in patients on ADT is a commonly overlooked and undertreated condition. Considering the long natural history of PCa, appropriate monitoring and management are essential in preserving bone health and health-related QoL in patients with PCa on long-term ADT. This article will review current guidelines available for the monitoring and management of ADT-induced bone loss in patients with PCa and will explore the effects of current lifestyle modifications and interventions, particularly on dietary supplementation and physical exercise in managing ADT-related alterations to bone health.

## 2. Pathophysiology Underlying Prostate Cancer Bone Metastasis

The skeletal system is the most common site of PCa metastasis, with 90% of patients with advanced PCa having bone involvement [8]. The most common sites are the vertebral column, pelvic bone, ribs, long bones, and the skull. Bone metastases are associated with SREs, such as pathological fracture, spinal cord compression, and surgical or radiation intervention of the bone. SREs are known to increase morbidity and to negatively impact QoL [9]. The aim of treatment for SREs is focused on delaying the onset of adverse events and maintaining QoL and functional status in patients with bone metastatic PCa [10].

The underlying pathogenesis of bone metastases in patients with PCa remains unclear, and studies are ongoing to elucidate the mechanisms of this entity. High levels of biochemical markers of bone turnover in patients with PCa with bone metastases indicate increased rates of osteoblast and osteoclast activity [11]. Moreover, researchers have hypothesized that interactions between the bone microenvironment and cancer cells may induce a vicious cycle of bone destruction and metastasis of the tumor [12]. Circulating tumor cells secrete interleukins, a gradient of chemokines, and parathyroid hormone-related protein (PTH-rP), which initiates osteolysis and results in the release of cytokines and factors by the bone marrow. The receptor activator of nuclear factor-ĸβ ligand (RANKL) is a mediator of bone remodeling and binds to its receptor on the surface of osteoclast progenitors, which leads to bone resorption. Tumor cells increase RANKL expression on osteoblasts by secreting PTH-rP, which leads to the proliferation of osteoclasts and increased bone resorption. In turn, osteoclast precursors, such as bone morphogenetic proteins, fibroblast growth factor, platelet-derived growth factor, and transforming growth factor-β, are activated. This stimulates tumor cell proliferation and the production of resorptive factors, completing the vicious cycle of tumor metastasis and osteolysis (Figure 1) [13]. Further studies focusing on the underlying pathophysiology directing the progression of bone metastases in patients with PCa would be crucial to broadening the horizon of novel therapeutic strategies [14].

## 3. Pathophysiology of Cancer Treatment-Induced Bone Loss

The progression of PCa is an androgen-dependent process. After the administration of ADT, sex steroid levels rapidly decrease and reach a nadir within four weeks of therapy [15]. The interactions between bone homeostasis and sex steroids have been widely explored [16,17,18,19]. In the skeletal system, androgens directly stimulate growth plate chondrocytes and influence the longitudinal growth of bone [20]. AR signaling in osteoblasts promotes differentiation and increases the protective effects of androgens on trabecular bone mass, which in turn decreases bone resorption and osteoclast numbers [18,21]. The role of estrogens in bone is to protect against endocortical resorption, which is regulated by estrogen receptors (ERα, ERβ) that are expressed by mesenchymal or stromal cells [22]. Estrogens play an essential role in osteoblast proliferation along with the inhibition of osteoclast precursors and in the regulation of osteoclast apoptosis [16,22,23]. Accumulating evidence supports the hypothesis that compared to testosterone, estrogen plays a more influential role in the regulation of bone metabolism in males [24,25,26].

Following the administration of ADT, serum levels of testosterone and estradiol show a substantial and rapid decrease, which leads to disruptions of skeletal homeostasis and bone rigidity. A prospective trial conducted by Greenspan et al. evaluated the rate of bone loss after ADT administration in patients with PCa and observed that the reduction in BMD was most rapid during the first year of ADT initiation, suggesting that preventive measures should be considered during this early period [15]. The duration of ADT and the harmful side effects on bone health show a direct relationship. In a separate cross-sectional study reported by Kiratli et al., a constant reduction in BMD was observed in patients with PCa who received long-term ADT. This effect was more prominent in patients who underwent continuous ADT and surgical castration in comparison to those who received intermittent ADT [27].

Bone loss induced by glucocorticoids is caused by enhanced rates of osteoblast and osteocyte apoptosis, impaired osteoblast differentiation, and prolonged osteoclast activity [28]. Glucocorticoids play a distinctive role in the management of CRPC, where they are used in combination with chemotherapy or administered as low dosage monotherapy [29,30]. Future investigations are warranted to explore management strategies to reduce the adverse effects of glucocorticoids on skeletal health when co-administered with ADT or adjuvant chemotherapy.

Androgen receptor axis targeted agents, including enzalutamide and abiraterone acetate in combination with prednisone, have gained FDA approval for their survival benefit in patients with advanced disease. In addition to their survival benefits, pooled analysis investigating the impact of both agents on skeletal endpoints has demonstrated an advantage in terms of SREs [31]. It is unclear whether this effect is secondary to the systemic control of bone metastasis or a direct effect on the bone microenvironment. In need of reducing the adverse metabolic effects of ADT, several studies have investigated the effect of these agents on bone metabolism. A single-arm, phase II trial investigated the potential of enzalutamide as a monotherapy for patients with hormone-naïve PCa eligible for ADT [32]. Bone turnover and metabolic outcomes were included in the exploratory outcomes, which indicated a stable BMD, with only small changes comparable to those achieved with bicalutamide and in contrast to those for leuprolide. Increased levels of estrogen and bone turnover markers of bone resorption with enzalutamide monotherapy were suggested as the pathophysiological mechanism of BMD maintenance [33]. Considering that enzalutamide is administered without the need for glucocorticoids, future trials would be warranted to investigate the long-term efficacy and safety of enzalutamide as monotherapy in terms of survival and cancer treatment-induced bone loss (CTIBL). For abiraterone acetate, the results from an in vitro study suggest a direct bone anabolic and an anti-resorptive effect [34]. Furthermore, abiraterone acetate was found to exhibit an inhibitory effect on human primary osteoclast function and to promote osteoblast differentiation and bone matrix deposition in patients with mCRPC [31]. Future studies are warranted to investigate how abiraterone acetate and enzalutamide may be utilized to exert additional positive effects on CTIBL due to ADT.

The increased adaptive exposure of ARs results in an incremental resistance to ADT. Evidence has demonstrated that this increase produces a therapeutic vulnerability, which leads to cellular apoptosis induced by supraphysiologic levels of androgens [35,36]. Bipolar androgen therapy has been proposed as an alternative therapeutic approach, in which rapid hormonal cycling between supraphysiologic and castrate levels of testosterone is performed to disrupt adaptive regulation of AR levels [37]. The BATMAN study was a phase II trial that investigated the feasibility of bipolar androgen therapy in terms of prostate-specific antigen response and metabolic effects in patients with hormone-sensitive PCa [38]. BMD was measured using dual-energy X-ray absorptiometry (DXA) at baseline and 6 and 15 months. As expected, there was no decline in BMD throughout the study period. Of note, the relatively small sample size and a short follow-up period of the study may have limited the ability to detect changes in bone metabolism.

Apalutamide is another competitive inhibitor of the AR. SPARTAN was a phase III trial that demonstrated prolonged metastatic-free survival and time to symptomatic progression in patients with non-metastatic CPRC [39]. Despite the survival benefit, the results indicate that treatment with apalutamide increased the risk of non-pathological bone fractures. The exact pathophysiology underlying non-pathological fractures is unclear; however, it may be presumable that more potent inhibition of testosterone activity may have accelerated bone turnover and subsequent fragility of the bone matrix.

## 4. Clinical Factors of Bone Loss

### 4.1. Primary Causes

When starting patients on ADT, it is necessary to consider risk factors that may cause bone loss in patients with PCa. According to Saylor et al., the risk of skeletal fractures from secondary causes other than osteoporosis is approximately 20% [40]. Cheung et al. showed that 11% of patients with PCa undergoing ADT harbored osteoporosis, while 40% of patients were diagnosed with osteopenia [41]. These reports indicate the importance of detailed medical evaluations for risk factors and potential underlying causes of bone loss prior to ADT administration.

During the aging process, the resorption–formation ratio increases in the skeletal system, which induces changes in bone density and strength [42]. Although increased periosteal circumference compensates for the rise in resorption rates, a shrinkage in cortical thickness causes the bone to become thinner and more porous, which reduces the strength of the bone matrix, and elicits a decline in BMD [43].

Endocrine factors play a crucial role in the age-associated changes in BMD. Studies have indicated that sex hormones, such as testosterone and the more potent estrogen, are significant inhibitors of bone resorption [44]. During the aging process, physical alterations occur on the cellular and systemic levels. Oxidative stress, cellular apoptosis, and autophagy are prime examples of the diverse pathophysiologies associated with vast alterations at the cellular level [44]. On the systemic level, increased serum levels of sex hormone-binding globulin reduce the bioavailability of hormones and the production of testosterone. Moreover, the adrenal gland production of dehydroepiandrosterone decreases along with the reduced production of luteinizing hormone and follicle-stimulating hormone, which reduces BMD in older adult men [45].

### 4.2. Secondary Causes

Osteoporosis caused by etiologies apart from age-related factors is denoted as secondary osteoporosis. Etiologies for secondary osteoporosis in males are presented in Table 1. Chronic kidney disease (CKD) is a significant risk factor for secondary osteoporosis. Patients with CKD suffer from vitamin D insufficiency and calcium malabsorption and exhibit increases in serum PTH levels, which are vital elements for healthy bone metabolism [45]. In older adult patients, reduced physical activity and protein intake, along with the aforementioned endocrine changes, are related to the degenerative reduction in skeletal muscle mass. A reduction in muscle mass causes increased secretion of bone promoting factors from muscles and decreased bone formation. Moreover, reduced muscle mass and strength expose older adults to falls and resultant bone fractures [43,46].

Bone metastasis in patients with PCa is a significant risk factor for secondary osteoporosis due to the altered structure of bone prior to ADT administration. Metastatic cancer cells overstimulate osteoclast and osteoblast activation, and the vicious cycle paradoxically decreases the integrity of the bone, since weaker woven bone is produced instead of lamellar bone. The continuous osteolytic cycle caused by osteoclasts leads to further fragility of the bone mineral matrix and results in a greater risk of pathological fractures [47,48].

## 5. Bone Health Assessment in Patients with Prostate Cancer

In 2016, guidelines for the assessment of bone health in patients with PCa were published as a joint venture by the European Society for Radiotherapy and Oncology, European Association of Urology (EAU), and the International Society of Geriatric Oncology [49]. Their recommendations state that patients with PCa initiating long-term ADT should be assessed with DXA and subsequently with a fracture risk assessment tool (FRAX) for the assessment of individual risk of fracture. DXA is most commonly used to assess BMD. Specific measurement locations for DXA scans include the proximal femur, pelvic brim or the femoral neck, and lumbar spine [50]. Assessments are noted as a T-score, and osteoporosis is defined as a T-score of ≥2.5 standard deviations below the mean value for young, healthy adults [51]. However, the sensitivity of DXA for predicting fractures occurring in patients with a non-osteoporotic BMD is low. Studies have shown that fractures are not uncommon in patients with low bone mass or non-osteoporotic BMD [52]. Therefore, physicians should take into account other factors that may increase the risk of fractures, including age, sex, prior fracture history, familial history, and other lifestyle aspects. Nonetheless, DXA is the standard tool for the assessment of BMD and is most widely used in clinical practice.

Based on data from prospectively studied population cohorts, the FRAX algorithm accounts for demographic data and medical history to improve assessments of individualized fracture risk [53]. FRAX is utilized mainly for patients aged older than 40 years and estimates the ten-year risk of hip and major osteoporotic fractures. Risk factors that are accounted for by FRAX include demographics, comorbidities, initial BMD, long-term use of corticosteroids, alcohol or tobacco intake, medical history of fractures, and familial history [54,55].

Q Fracture is another diagnostic modality for the evaluation of bone health that was developed and validated using a cohort of over two million British patients [55,56]. Trabecular bone score (TBS) is another diagnostic algorithm for the evaluation of bone density in the lumbar spine. TBS utilizes a textural index according to pixel grey-level variations in DXA scans and is an indirect representation of bone architecture that can be used to monitor bone quality and to assess the fracture risk independent of BMD [57]. This diagnostic tool could be used for better assessment of the risk of fracture in patients with CTIBL. Moreover, it can potentially be utilized as an adjunct diagnostic modality when used in combination with FRAX and BMD to optimize the identification of high-risk patients [58]. Since it has not been validated in patients with PCa, there are no validated recommendations or guidelines for its routine use in clinical settings.

## 6. Monitoring Bone Health and Cancer Treatment-Induced Bone Loss in Patients with Prostate Cancer Treated by Androgen-Deprivation Therapy

In men undergoing ADT, the preservation of bone health is a crucial component in the prevention of fractures. The PCa population is itself susceptible to fractures owing to the side effects of ADT on BMD. A continuous prolongation in the life expectancy of patients during or after the treatment also poses increased harm to bone health due to the aging process, increased risk of falls related to neurological deficits, and progressive weakening of the muscles.

The EAU guidelines suggest that the interval of BMD analysis should be based according to baseline T scores before ADT administration [59]. DXA scanning is recommended to be annually repeated if the baseline T score lies between −2.5 and −1.0, or every two years if the baseline T score lies greater than −1.0. The International Society for Clinical Densitometry recommends annual assessment of BMD following the initiation of ADT, based on clinical changes in patients undergoing ADT [60]. Current screening guidelines for male osteoporosis from the World Health Organization and the National Osteoporosis Foundation advocate routine DXA scans in male patients above the age of 70 years. Men with risk factors, such as a history of prior fracture, family history of osteoporosis, low body mass index, tobacco use, alcoholism, corticosteroid usage, medical comorbidities, and low levels of vitamin D, are recommended to be tested between the ages of 50 to 69 years [61].

According to recent expert opinion, evaluations of bone alkaline phosphatase, vitamin D, serum calcium, and PTH levels should be performed at baseline and then every 12–18 months afterward to monitor ADT-induced adverse bone effects [54]. Moreover, the study emphasized that it is important not to overlook back pain and height loss, and to perform spinal radiographical examinations to early identify vertebral fractures [54].

Various guidelines and expert opinions for monitoring bone health in patients with PCa maintain a similar stance, such as taking a thorough medical history, screening of risk factors along with lifestyle modifications. However, the rate and frequency of bone health monitoring in patients with PCa are prone to changes and rely upon the clinical judgment and discretion of the physician, the patient, and practical factors. Therefore, a comprehensive set of guidelines or recommendations is required to guide the baseline and follow-up monitoring of bone health in patients with PCa on ADT. A summary of the clinical evaluation and monitoring recommendations for ADT-induced skeletal adverse events is provided in Table 2.

## 7. Management Strategies for Cancer Treatment-Induced Bone Loss

Osteoporosis and osteopenia are common underlying medical conditions before ADT administration, especially in older adult patients with PCa. Studies have identified factors that promote the preservation of bone health in this patient subset, such as maintaining adequate body mass, weight-bearing exercise, avoiding tobacco and alcohol, and high dietary calcium intake [62,63,64,65]. Bone-targeted therapies, such as bisphosphonates, human monoclonal antibodies, and selective estrogen receptor modulators, have proven effective for the prevention of BMD and the mitigation of SREs from PCa bone metastases. These therapeutic options have been extensively reviewed in the literature and are beyond the scope of this review [66]. Due to advances in treatment, the prolonged natural history of patients with bone-metastatic PCa has increased patients suffering from the impact of bone loss. Therefore, management and treatment strategies are essential for the effective prevention and treatment of CTIBL.

### 7.1. Awareness and Education

Previous reports have suggested that patients lack adequate awareness regarding the risks of CTIBL, prevention measures, and available treatment options [67,68,69]. In a study that explored communication and information exchange between physicians and patients about CTIBL and bone metastases, there were inconsistencies between what physicians assumed that patients are aware of and the perceptions and amount of information about bone health that patients actually acquired [70]. Even though clinicians are the primary sources of information on bone health, they often do not strictly follow guidelines on the screening, monitoring, and management of CTIBL [71,72]. A recent phase II study that evaluated two education-based models showed that the combined use of patient information pamphlets, along with active participation of clinicians and coordinators of bone health care, might improve the overall management of bone health for patients with PCa on ADT [73]. Considering that online information is one of the most utilized sources for patients, des Bordes et al. explored a systematic and transparent approach involving approved online educational tools or websites and the benefits of online education based on bone health. Their results indicate that PCa and breast cancer survivors could benefit from a validated online educational system for raising bone health awareness and improving knowledge of healthy bone behavior [74]. Overall, a systematic model is warranted for the promotion of awareness and education on bone health for patients with bone metastases.

### 7.2. Dietary Supplementation

Based on their potential roles in the mitigation of ADT-induced adverse bone loss and osteoporosis, calcium and vitamin D supplements for patients with PCa have been three topic of numerous reports [75,76,77,78]. Supplemental consumption of calcium and vitamin D is endorsed, and the recommended dosage intakes vary from 1000 to 1500 mg/day and from 800 to 2000 IU/day, respectively [6,63,64,79]. For men with vitamin D deficiency, the recommended supplement dosage is 3000–5000 IU/day of vitamin D for at least 6–12 weeks under clinical guidance depending on the level of deficiency [80]. These guidelines are in line with the results of a critical review performed by Datta and Schwartz, which concluded that a supplementary dosage of 500–1000 mg calcium and 200–500 IU vitamin D/day was unable to prevent bone loss in men treated with ADT [77].

A meta-analysis suggested that increasing the intake of calcium or dairy consumption may increase the risk of PCa; however, the study also concluded that supplemental calcium was not associated with a higher risk of PCa [81]. Other similar studies have also shown that supplemental calcium at doses of 1500 mg per day was not associated with PCa progression. Likewise, dietary intake of at 4000 IU of vitamin D per day for one year in patients with PCa showed that there was no increased risk of adverse effects or disease progression [82]. Further studies are warranted to elucidate the risk and benefits of calcium and vitamin D supplementation on bone health in patients with PCa receiving ADT. However, based on current clinical evidence, the primary physician or urologist should consider calcium and vitamin D supplementation as a key component in their treatment strategy for men undergoing ADT (Table 3).

### 7.3. Lifestyle Modification and Physical Exercise

The cessation of smoking, moderate sun exposure, and limited alcohol intake of less than two standard drinks (one standard drink contains 14 g or 0.6 fluid ounces of pure alcohol) per day are non-pharmacological interventions that can be recommended to patients on ADT [63,65]. Sarcopenia is a degenerative loss of muscle mass closely associated with physical frailty, which increases the risk of falls and subsequent risk of skeletal fractures [83]. When sarcopenic patients are exposed to ADT, the risk of fracture increases along with its potentially life-threatening complications [84]. Countermeasures for ADT-induced sarcopenia should include muscle-strengthening exercise and evaluation for fall prevention. Such evaluations include neurological examination and a thorough assessment of prescription medications for side effects that could affect balance [85]. These general lifestyle modifications have been shown to potentially prevent sarcopenia and its associated consequences in patients on ADT [86].

Physical exercise for the prevention of ADT-induced osteoporosis is a widely recommended treatment modality. Suggested exercise regimens include weight-bearing training, progressive resistance training (PRT), and balance training [64,65]. However, these guidelines lack specific exercise prescriptions, such as training interval, intensity, and duration. A 20-week PRT program was conducted in ten patients on ADT, which consisted of exercises that started from two sets at 12 repetitions maximum to four sets at six repetitions maximum [87]. However, there were no improvements in BMD scores assessed at the femoral neck, trochanter, or Ward’s triangle. Nilsen et al. observed changes in the femoral neck, trochanter, total hip, or lumbar spine BMD in 28 patients on ADT who underwent a 16 week PRT regimen [88]. The results show meaningful improvements in lean body mass in the lower and upper extremities; however, no changes were observed regarding areal BMD measured by DXA. Cormie et al. [89] described similar results. The exercise regimen consisted of eight exercises, which progressed from one to four sets of six to 12 repetitions maximum, performed twice every week. After a 12-week PRT and weight-bearing regimen, the ADT group showed a 0.91% loss in lumbar spine BMD and a 0.12% loss in hip BMD, compared to the control group. However, the results were not statistically significant. The limitations of the studies mentioned above were the limited sizes of study samples and short study periods, which precluded demonstrating clinically relevant and beneficial effects on BMD. Therefore, clinical trials with larger study samples and longer-term follow-up periods are warranted.

A longer-term follow-up trial analyzed the role of physical exercise on BMD in patients with PCa undergoing ADT. The study cohort was subjected to a weight-bearing and PRT physical program, which lasted for 12 months. There were no clinically significant differences in bone density in patients on ADT, compared to the exercise placebo group, in which the exercise regimen consisted of stretching. Compared to controls, the difference in lumbar spine BMD was 0.92%, femoral neck BMD decreased by 0.97%, and total hip BMD slightly increased by 0.07% in the study group. However, these results are not statistically significant [90]. The authors noted that limited sample size and low adherence to home-based training were potential limitations to their ability to elicit improvements in bone health in these participants.

A Danish group performed a randomized trial investigating the effect of recreational football intervention on BMD in patients with PCa on ADT [91]. This study was the only exercise modality that resulted in statistically significant increases in BMD. Patients that participated in the football exercise showed a 2.1% difference at the lumbar spine (*p* = 0.144), a 1.7% difference at the femoral neck (*p* = 0.078), and clinically significant 1.7% difference at the total hip (*p* = 0.015) [91]. The results of this study infer that exercise training could be recommended as a crucial adjuvant to pharmacological treatments for the care and prevention of CTIBL in patients with PCa on ADT. A study based on 180 older adult men with osteoporosis showed that a combined work out of moderate-to-high-intensity PRT with weight-bearing impact exercise sessions three times per week was effective for increasing femoral neck and lumbar spine BMD over 12 months, compared with non-exercising controls [92]. Similarly, a meta-analysis reported that exercise programs consisting of moderate-to-high-intensity PRT and weight-bearing exercises were most efficacious in maintaining and improving the BMD of the hip and spine [93].

A randomized controlled study investigated the impact of structured lifestyle interventions on device-measured physical activity and QoL in patients with PCa receiving ADT [94]. Patients on long-term ADT were randomized to the intervention arm, which received a structured lifestyle intervention, including text messages from sports medicine specialists for eight weeks. At each visit, self-reports and accelerometers were utilized to evaluate physical activity and sedentary behavior, and questionnaires were used to measure QoL, life satisfaction, anxiety, and depression. Significantly greater improvements in QoL (*p* = 0.005) and depression (*p* = 0.001) compared to baseline were reported in the intervention group compared to the control group. Although this study did not investigate the efficacy of interventional physical exercise on bone health, the results implied that a supervised lifestyle intervention program should be offered to patients on ADT for its advantage in improving QoL and reducing depression.

Current clinical trials regarding lifestyle modification and physical exercise have been usually conducted with small study populations and have a short follow-up period. Moreover, the clinical endpoints and the tools for measurement in each study preclude a quantitative analysis of how the interventions can prevent bone loss and preserve QoL. Further prospective and long term studies are warranted to evaluate the efficacy and safety of physical exercise on bone health in patients with PCa. For now, based on the literature and previous research, we encourage patients with PCa who are treated with ADT to engage in an exercise program under the supervision of a qualified exercise physiologist or trainer (Table 3).

## 8. Conclusions

ADT has become a cornerstone in the management of locally advanced and metastatic PCa. Although ADT has improved overall survival in this select group of patients, adverse effects include loss of BMD, a potential risk factor of osteoporosis, falls, and fractures. Monitoring and maintenance of bone health in patients undergoing ADT should be a clinical goal across the treatment landscape of PCa. Despite variation in contemporary guidelines for PCa treatment, there is an unmet need to evaluate and implement the growing amount of evidence supporting non-pharmacological approaches such as lifestyle interventions and physical training, to mitigate and manage ADT-induced adverse skeletal events. Future prospective trials are warranted to validate the clinical benefits and potential risks of lifestyle modifications and to define an optimal physical exercise regimen for the improvement of bone health in this population.

## Figures and Tables

**Figure 1 cancers-12-01529-f001:**
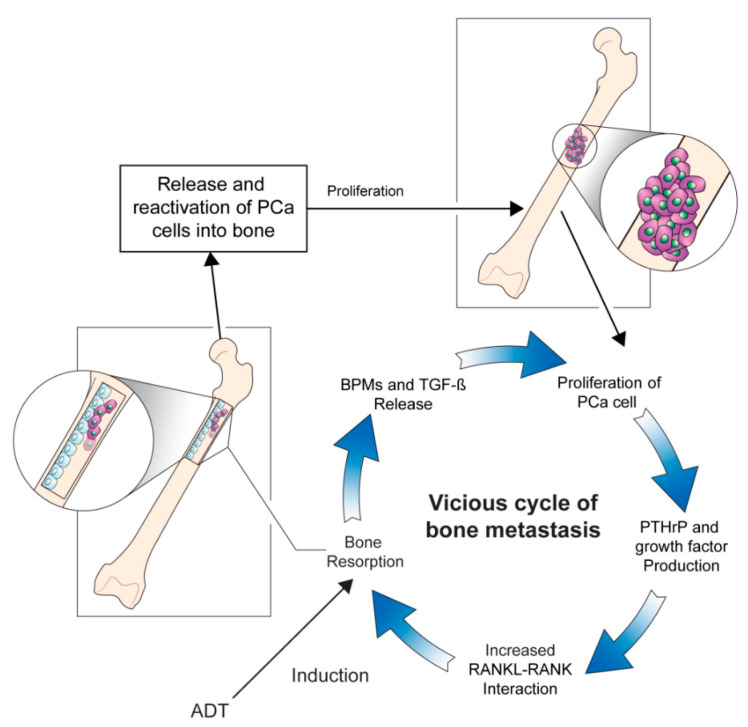
The vicious cycle of bone metastasis. Bone resorption releases and reactivates PCa cells into the bone, leading to metastatic outgrowth in the bone microenvironment. Production of factors by PCa cells that increase bone resorption through enhanced interaction between RANKL expressing osteoblasts and RANK expressing osteoclasts mediates the vicious cycle of PCa bone metastasis. Production of PTHrP and growth factors increases the interaction between RANKL-expressing osteoblasts and RANK-expressing osteoclasts, which further increases bone resorption. The release of BMPs and TGF-β by bone resorption further aggravates the vicious cycle. ADT: androgen-deprivation therapy, BMPs: bone morphogenetic proteins, PCa: prostate cancer, PTHrP: parathyroid hormone-related protein, RANK: receptor activator of nuclear factor-ĸβ, RANKL: receptor activator of nuclear factor-ĸβ ligand, TGF-β: transforming growth factor-β.

**Table 1 cancers-12-01529-t001:** Causes of osteoporosis in men, excluding drugs.

**Endocrine**
Chronic kidney disease Diabetes mellitus
Hyperparathyroidism
Hypercortisolism
Hypogonadism, including aging
Hyperthyroidism
**Nutritional/Gastrointestinal**
Alcoholism
Chronic liver disease
Inflammatory bowel disease
Malabsorption syndromes
Malnutrition
**Rheumatological/Connective Tissue**
Ankylosing spondylitis
Marfan syndrome
Rheumatoid arthritis
Systemic lupus erythematosus
**Hematological**
Disseminated bone metastasis
Lymphoma/Leukemia
Multiple myeloma

**Table 2 cancers-12-01529-t002:** Summary of clinical guidelines for the assessment and monitoring of adverse skeletal events.

**At the Initiation of ADT** **Evaluation for Any History of Trauma-Induced Fractures and Risk Factors for Osteoporosis [49]**
BMD assessment with DXA scan and subsequent scoring with FRAX [49]
DXA scan or FRAX score only is not recommended. The following factors should be incorporated [49,53,54]
Age
BMD
History of corticosteroid therapy
Medical history of bone metastasis or fragility disease or treatment
Physical disability or risk factors of fall
**Evaluations Recommended for Monitoring Skeletal Health (Perform at Baseline and Every 12–18 months Afterward) [54,59]**
BMD measurement using DXA scan during the first 24 months of ADT
Bone turnover markers (e.g., serum ALP level)
Serum calcium levels
Serum vitamin D levels
Serum PTH levels
Height, weight, BMI
In the case of lumbar pain or loss of height, perform spine radiography and imaging studies

ADT: androgen-deprivation therapy; ALP: alkaline phosphatase; BMD: bone mineral density; BMI: body mass index; DXA: dual-energy X-ray absorptiometry; FRAX: fracture risk assessment tool; PTH: parathyroid hormone.

**Table 3 cancers-12-01529-t003:** Lifestyle modifications and physical training recommendations for managing androgen-deprivation therapy-related adverse events.

**Tobacco and Alcohol Consumption**
Smoking cessation should be recommended [62,63]
Alcohol consumption should be limited to two or fewer standard drinks per day [63]
**Calcium**
Integrate 3–4 daily servings of dairy products for dietary calcium intake [6,63,64,75,76,77,78]
Consider calcium supplements if the daily calcium intake is below 1000–1300 mg per day [6,63,64,75,79]
**Vitamin D**
Prior to ADT initiation, patients should have an assessment of serum 25(OH)D [6,63,74,75,77,79]
Men with 25(OH)D levels ≥50 nmol L^−1^ should consider a daily supplement intake of 800 IU. [6,63,64,75,77,79]
Men with vitamin D deficiency should supplement with 3000–5000 IU per day for at least 6–12 weeks under clinical supervision [75,76,77,78,80]
****Physical Exercise Training****
Progressive overloading should be applied to all exercises when possible, and sessions must be performed under professional supervision [64,65]
Progressive resistance training (PRT) [87,88,89,90,91]
Consider at least two times per week
At least 10 exercises focusing on major muscle groups, especially the muscles attached to the lumbar spine and hip
Two to three sets of 8–10 repetitions at moderate to high intensity
Weight-bearing impact exercise [89,90,91,92,93]
At least 4 days per week
Two to four impact exercises with a variation in magnitude and duration
PRT is recommended to patients with low muscle strength before initiation of impact exercises
Aerobic training [65]
At least five times per week
At least 30 min of continuous exercise
Target heart rate should be 55–75% of maximum predicted heart rate
Training sessions can be divided into shorter sessions if needed (three individual 10-min sessions)

25 (OH)D: 25-hydroxyvitamin D; PRT: progressive resistance training.

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
