# Peer review of "Pathophysiology of Bone Loss in Patients with Prostate Cancer Receiving Androgen-Deprivation Therapy and Lifestyle Modifications for the Management of Bone Health: A Comprehensive Review"

_cancers, 2020, doi:10.3390/cancers12061529_

Round 1

Reviewer 1 Report

Bone is the most common organ site where Prostate Cancer metastasizes to, and bone pain is a common symptom in patients with metastatic disease. In this review, the authors gave an overview of various factors causing bone loss in PCa patients, and provide a list of strategies for managing cancer treatment-induced bone loss. This review has direct clinical relevance for guiding PCa patients to preserve bone health.

Meanwhile, there are also some concerns/suggestions for this review:

1)PCa that develops bone metastasis normally has undergone extensive ADT, and will receive more potent AR antagonists for treatment instead, such as Enzalutamide or Abiraterone. This second-line treatment should also be mentioned in the introduction.

2)A simple cartoon illustrating the vicious cycle of bone loss can be very helpful for readers.

3)Information for "Author Contribution" "Funding" etc is missing.

Author Response

Response to review comments (reviewer #1)

Bone is the most common organ site where Prostate Cancer metastasizes to, and bone pain is a common symptom in patients with metastatic disease. In this review, the authors gave an overview of various factors causing bone loss in PCa patients, and provide a list of strategies for managing cancer treatment-induced bone loss. This review has direct clinical relevance for guiding PCa patients to preserve bone health.

REPLY: We are very much thankful to the reviewer for the thorough review. We agree with all specific comments raised and have revised our paper in light of the useful suggestions. Answers to the specific comments/suggestions/queries are provided below.

1. PCa that develops bone metastasis normally has undergone extensive ADT, and will receive more potent AR antagonists for treatment instead, such as Enzalutamide or Abiraterone. This second-line treatment should also be mentioned in the introduction.

REPLY: Indeed, during the last decade, androgen receptor axis-targeted (ARAT) agents such as enzalutamide and abiraterone, have been approved for the treatment of mCRPC patients. As the reviewer has pointed out, trials, such as PROSPER and LATITUDE, have demonstrated survival benefit of ARAT agents and broadened the clinical indications. In line with the reviewer’s comment, we have added information on the aforementioned agents in the Introduction section (lines 42-46). However, we would be grateful if the reviewer may understand that this invited review article was to mainly focus on lifestyle interventions for the management of bone health. Therefore, the added sentences were kept brief. Thank you again for your insight.

2. A simple cartoon illustrating the vicious cycle of bone loss can be very helpful for readers.

REPLY: Thank you for this valuable comment. We have added a figure (Fig. 1) illustrating the vicious cycle of bone metastasis (lines 85-97). We believe that our diagram will show how bone resorption induced by ADT may lead to the proliferation of PCa cells and subsequent production of PTHGrP and growth factors and increased interactions of RANKL and RANK.

3. Information for "Author Contribution" "Funding" etc is missing.

REPLY: Thank you for pointing this out. Information on “Author Contribution,” “Funding,” and “Conflicts of interest” were added to the manuscript.

Reviewer 2 Report

This is a review paper titled Lifestyle modifications for the management of bone health in patients with prostate cancer receiving androgen-deprivation therapy. However, only a small section at the end of manuscript discussing lifestyle modification.  The rest of the paper is a summary of the bone related diseases associated with ADT or prostate cancer.

The paper will be benefit from focusing on lifestyle factors and including how intervention or modification will prevent or modify bone related diseases.  Also, a quantitative analysis such as comparing results from previous studies showing how much lifestyle modification can prevent bone loss and keep quality of life will increase the level of contribution of their work to the field.

Minor suggestion

In Table 3, references will be needed.

Author Response

Response to review comments (reviewer #2)

We are very much thankful to the reviewer for the thorough review. We agree with all specific comments raised and have revised our paper in light of the useful suggestions. Answers to the specific comments/suggestions/queries are provided below.

1. This is a review paper titled Lifestyle modifications for the management of bone health in patients with prostate cancer receiving androgen-deprivation therapy. However, only a small section at the end of manuscript discussing lifestyle modification. The rest of the paper is a summary of the bone related diseases associated with ADT or prostate cancer.

REPLY: Thank you for this comment. We are very much aware that the main scope of this review is on lifestyle modifications for bone health in patients with prostate cancer. However, available clinical evidence and guidelines individually based on bone health management was limited. We would be grateful if the reviewer may understand that we thoroughly and extensively reviewed all currently available literature and tried to reflect all relevant information as a comprehensive review. For additional clarity on this issue, we have edited the title of our article.

2. The paper will be benefit from focusing on lifestyle factors and including how intervention or modification will prevent or modify bone related diseases. Also, a quantitative analysis such as comparing results from previous studies showing how much lifestyle modification can prevent bone loss and keep quality of life will increase the level of contribution of their work to the field.

REPLY: Thank you for this insightful comment. The limitation of this topic is that currently, there are no solid and unified guidelines for lifestyle modifications concerning bone health preservation. Various recommendation and expert opinions for monitoring and management of bone health in patients with prostate cancer maintain a similar position, such as taking a thorough medical history, screening of risk factors along with lifestyle modifications. However, the rate and frequency of bone health monitoring in patients with PCa are prone to changes and rely upon the clinical judgment and discretion of the physician, the patient, and practical factors. We have also noted that current clinical trials regarding lifestyle modification and physical exercise have been usually conducted with small study populations and have a short follow-up period. Moreover, the clinical endpoints and the tools for measurement in each study preclude a quantitative analysis on how the interventions can prevent bone loss and preserve QoL. According to the reviewer’s comment, we have added this limitation to the Discussion section (lines 387-390).

3. In Table 3, references will be needed.

REPLY: Thank you for pointing this out. We have added the relevant references for the information listed in Table 3.

Reviewer 3 Report

The review by Kim and Koo describes in a comprehensive manner the correlation between lifestyle and bone mass density in prostate cancer patients.

The reduced BMD by androgen deprivation therapy has been reviewed but the association with life style is an important aspect that should be considered in the medical care.

Some points should be considered prior publication:

  1. Androgen deprivation therapy is induced by GnRH agonists or antagonists. However, most patients receive in addition for full AR blockade AR antagonists. Authors might speculate on BMD whether ADT alone or in combination with AR antagonists differ.
  1. An important aspect is that most elderly prostate cancer patients have a reduced BMD prior ADT. Authors should discuss the decline of androgens by age as a risk factor for reduced BMD. This includes also adding this factor in table 2
  2. Unclear is in table 2 the definition of excessive exercise.
  3. Please define: line 275 and table 3: limited alcohol consumption of less than two standard drinks per day. What exactly is meant by two standard drinks?
  4. Authors miss to mention the bipolar androgen therapy using cycles of supraphysiological androgens and discuss about BMD in an outlook.
  5. Authors should also speculate on selective AR antagonists that may differ in their action on BMD in an outlook.

Author Response

Response to review comments (reviewer #3)

The review by Kim and Koo describes in a comprehensive manner the correlation between lifestyle and bone mass density in prostate cancer patients. The reduced BMD by androgen deprivation therapy has been reviewed but the association with life style is an important aspect that should be considered in the medical care.

REPLY: We are very much thankful to the reviewer for the thorough review. We agree with all specific comments raised and have revised our paper in light of the useful suggestions. Answers to the specific comments/suggestions/queries are provided below.

1. Androgen deprivation therapy is induced by GnRH agonists or antagonists. However, most patients receive in addition for full AR blockade AR antagonists. Authors might speculate on BMD whether ADT alone or in combination with AR antagonists differ.

REPLY: Thank you for this insightful comment. We agree that this important information was neglected. In addition to the survival benefits of androgen receptor axis-targeted (ARAT) agents, pooled analysis investigating the impact of these agents on skeletal endpoints has demonstrated an advantage in terms of bone metabolism. The results of a phase II study investigating the effect of enzalutamide monotherapy for patients with hormone-naïve PCa eligible for ADT indicated that enzalutamide provides a stable BMD, with only small changes comparable to that of bicalutamide and in contrast to that of leuprolide. The authors suggested increased levels of estrogen and bone turnover markers of bone resorption with enzalutamude monotherpay as the pathophysiological mechanism of BMD maintenance. For abiraterone acetate, results from an in vitro study have suggested a direct bone anabolic and an anti-resorptive effect. Furthermore, abiraterone acetate was observed to exhibit an inhibitory effect on human primary osteoclast function and to promote osteoblast differentiation and bone matrix deposition in patients with mCRPC. This information was added to the manuscript (lines 128-148). Thank you again for this comment.

2. An important aspect is that most elderly prostate cancer patients have a reduced BMD prior ADT. Authors should discuss the decline of androgens by age as a risk factor for reduced BMD. This includes also adding this factor in table 2

REPLY: We absolutely agree with the reviewer’s comment. In clinical practice, since bone fragility may be present in patients prior to the initiation of ADT and throughout the disease continuum, close attention should be paid to bone health. The early onset of fractures should be taken into account when managing fracture risk and treatment timing. Moreover, it would be important to plan strategies to prevent, assess, and treat both osteoporosis and associated risk of fractures related to the aging process. According to the reviewer’s comment, we have subdivided Section 4 into separate subsections (primary and secondary causes) for reading fluidity. We have also added ‘aging’ as a risk factor in Table 1. Thank you for pointing this out.

3. Unclear is in table 2 the definition of excessive exercise.

REPLY: We agree with the reviewer that ‘excessive exercise‘ is an ambiguous term, and we deleted it from the table. We have further edited the table and rearranged the contents for clarification. Thank you for this comment.

4. Please define: line 275 and table 3: limited alcohol consumption of less than two standard drinks per day. What exactly is meant by two standard drinks?

REPLY: We agree with the reviewer that the definition of ‘standard drinks’ needs to be described. According to our references, a standard drink is any drink that contains about 0.6 fluid ounces or 14 grams of pure alcohol. We have added this information to Section 7.3. (lines 325-327).

5. Authors miss to mention the bipolar androgen therapy using cycles of supraphysiological androgens and discuss about BMD in an outlook.

REPLY: Thank you for this insightful comment. We agree that bipolar androgen therapy (BAT) can be regarded as an option to ameliorate adverse metabolic effects of ADT on BMD, and should have been discussed. A recent phase II trial (BATMAN) investigated the feasibility of BAT regarding PSA response and metabolic effects in patients with hormone-sensitive PCa. BMD was measured using dual-energy X-ray absorptiometry at baseline and 6 and 15 months. As expected, there was no decline in BMD throughout the study period. Of note, the relatively small sample size and a short follow-up period of the study may have limited the ability to detect changes in bone metabolism. We have added this information to Section 3 (lines 149-159).

6. Authors should also speculate on selective AR antagonists that may differ in their action on BMD in an outlook.

REPLY: Thank you for this comment. Evidence from the SPARTAN trial shows that treatment with novel AR antagonists increases the risk of bone fracture in patients with CRPC undergoing long-term ADT. Further clinical studies would be needed to elucidate the exact mechanism for BMD loss; however, it may be presumable that more potent inhibition of testosterone activity may have accelerated bone turnover and subsequent fragility of the bone matrix. This information was added to Section 3 (lines 160-166).

Round 2

Reviewer 2 Report

Previous comments have been addressed.